# Collagen-like Osteoclast-Associated Receptor (OSCAR)-Binding Motifs Show a Co-Stimulatory Effect on Osteoclastogenesis in a Peptide Hydrogel System

**DOI:** 10.3390/ijms25010445

**Published:** 2023-12-28

**Authors:** Mattia Vitale, Cosimo Ligorio, Stephen M. Richardson, Judith A. Hoyland, Jordi Bella

**Affiliations:** Division of Cell Matrix Biology and Regenerative Medicine, School of Biological Sciences, Faculty of Biology, Medicine and Health, The University of Manchester, Manchester M13 9PT, UK; mattia.vitale@bivictrix.com (M.V.); cosimo.ligorio1@nottingham.ac.uk (C.L.); s.richardson@manchester.ac.uk (S.M.R.); judith.a.hoyland@manchester.ac.uk (J.A.H.)

**Keywords:** osteoclast, OSCAR receptor, osteoclastogenesis, bioactive motifs, peptide hydrogel

## Abstract

Osteoclastogenesis, one of the dynamic pathways underlying bone remodelling, is a complex process that includes many stages. This complexity, while offering a wealth of therapeutic opportunities, represents a substantial challenge in unravelling the underlying mechanisms. As such, there is a high demand for robust model systems to understand osteoclastogenesis. Hydrogels seeded with osteoclast precursors and decorated with peptides or proteins mimicking bone’s extracellular matrix could provide a useful synthetic tool to study pre-osteoclast-matrix interactions and their effect on osteoclastogenesis. For instance, fibrillar collagens have been shown to provide a co-stimulatory pathway for osteoclastogenesis through interaction with the osteoclast-associated receptor (OSCAR), a regulator of osteoclastogenesis expressed on the surface of pre-osteoclast cells. Based on this rationale, here we design two OSCAR-binding peptides and one recombinant OSCAR-binding protein, and we combine them with peptide-based hydrogels to study their effect on osteoclastogenesis. The OSCAR-binding peptides adopt the collagen triple-helical conformation and interact with OSCAR, as shown by circular dichroism spectropolarimetry and surface plasmon resonance. Furthermore, they have a positive effect on osteoclastogenesis, as demonstrated by appropriate gene expression and tartrate-resistant acid phosphatase staining typical of osteoclast formation. Combination of the OSCAR-binding peptides or the OSCAR-binding recombinant protein with peptide-based hydrogels enhances osteoclast differentiation when compared to the non-modified hydrogels, as demonstrated by multi-nucleation and by F-actin staining showing a characteristic osteoclast-like morphology. We envisage that these hydrogels could be used as a platform to study osteoclastogenesis and, in particular, to investigate the effect of costimulatory pathways involving OSCAR.

## 1. Introduction

Receptor activator of nuclear factor kappa beta (RANK) is well-known to have a key effect on the differentiation of osteoclast precursor cells (OCPs) into osteoclasts (OCs) through binding to RANK ligand (RANKL) [1,2,3]. Alongside the canonical RANKL/RANK pathway, OC differentiation requires the presence of co-stimulatory receptor signalling through the activation of adaptors that contain immunoreceptor tyrosine-based activation motifs (ITAMs), such as the Fc receptor common γ-chain (FcRγ). However, little is known about the activation of ITAM-containing receptors and their ligands [3,4,5]. The osteoclast-associated receptor (OSCAR) is a collagen receptor from the immunoglobulin superfamily that is expressed by OCs, chondrocytes, and other immune cells [6,7]. It has been hypothesised that the osteoclast-associated receptor (OSCAR) is involved during OCP differentiation as a co-stimulatory pathway. Further supporting this hypothesis, its predominant detection in bone and interaction with osteoblast (OB)-released factors propose a mechanism where OSCAR promotes OCP differentiation via FCRγ, alongside the RANKL/RANK pathway [2,3,4,5,6,7,8].

Nevertheless, the true nature of potential OSCAR ligands is largely unknown. It was found that OSCAR may interact with fibrillar collagens released in the extracellular matrix (ECM) from bone surfaces, where OCPs undergo maturation and terminal differentiation in vivo [5]. It is also known that OSCAR-collagen interactions play an important role during bone development, maintenance, repair, and disease [7,9,10,11]. For this reason, different research groups have tried to elucidate the OSCAR-collagen interaction, with particular attention to its capability to promote OC differentiation and osteoclastogenesis as well as mature OC activation [5].

The role of OSCAR in osteoclastogenesis and bone disease has been recently reviewed [4]. The molecular details of the interaction between OSCAR and fibrillar collagens have been investigated using collagen-like peptide libraries (toolkits) [5]. These studies have identified a minimal OSCAR-binding collagen sequence as GXOGPX’GFX’, where repeating GXX’ triplets represent the characteristic sequence of collagen proteins and O is the abbreviation for the amino acid 4-hydroxyproline (Hyp), typical of animal collagens [12]. The main fibrillar collagen types I, II, and III contain several instances of this OSCAR-binding sequence motif [5]. However, a complete description of the residue preferences at each X or X’ position has not been reported yet. Alanine scan mutations of this minimal binding sequence show that the Hyp residue at position 3 and the phenylalanine (F, Phe) residue at position 8 are required for OSCAR binding [5]. Crystal structures of the extracellular region of OSCAR in complex with short collagen-like triple-helical peptides containing an OSCAR-binding sequence provided a high-resolution structural view of the OSCAR-collagen interaction [9,13]. The structures show why the triple-helical conformation of collagen is critical for OSCAR binding and confirm the key role of the conserved residues in the OSCAR-binding collagen sequences.

Biomaterials mimicking bone-like microenvironments in vitro can be used to address specific questions on OSCAR-collagen interactions and their role in osteoclastogenesis. To this extent, self-assembling peptide hydrogels offer a fully synthetic, physico-chemically tuneable platform for cell culture, drug screening, and tissue engineering applications [14,15,16,17]. Moreover, bioactive peptides mimicking ECM proteins have been shown to provide signals to cells and to steer cell fate in multiple peptide-based systems [18,19]. In our previous work, we formulated bone-like hydrogel microenvironments by using fluorenylmethoxycarbonyl-diphenylalanine/serine (Fmoc-FF/S) hydrogels decorated with the fibronectin-derived motif arginine-glycine-aspartate (RGD) and with hydroxyapatite (Hap) nanoparticles. This composite hydrogel induces successful osteoclastogenesis in Raw 264.7 cells [20]. We also developed a simple, cost-effective ‘incorporation protocol’ to decorate Fmoc-FF/S hydrogels with full-length collagens, collagen-like peptides, or recombinant collagen-like proteins for the design of multifunctional hydrogel platforms [21].

In this study, we aimed to use our protocol to combine collagen-like peptides or proteins containing OSCAR-binding sequences GXOGPX’GFX’ [5] with our previously developed Hap-decorated and RGD-functionalised Fmoc-FF/S hydrogels. These multifunctional composite hydrogels, containing cell signalling motifs (i.e., integrin-binding and OSCAR-binding motifs) as bone-mimicking biological cues, as well as hydroxyapatite particles as mechanical cues, could be valuable in investigating the effect of the OSCAR-collagen interaction on the differentiation in vitro of Raw 264.7 cells. However, recombinant collagen proteins produced in bacteria do not contain Hyp; therefore, we first wanted to investigate whether replacing Hyp with proline (Pro) at the third position of the consensus sequence GXOGPX’GFX’ would have a significant effect on OSCAR binding. Based on this, we designed two synthetic collagen-like peptides containing the same OSCAR-binding motif, one with Hyp at the third position (named OP1_Hyp) and one with Pro (named OP1_Pro), and quantified their binding affinity to immobilised OSCAR. Then, a new recombinant collagen-like protein containing an OSCAR-binding motif (named OCol1) was engineered in our laboratory using a trimerization domain (PfC) from bacteriophage collagen-like proteins [22]. The two peptides and the recombinant protein were characterised biochemically and incorporated into a Hap-decorated Fmoc-FF/S/RGD hydrogel using our published protocol [21]. This multifunctional composite hydrogel was used to investigate the differentiation of Raw 264.7 cells into OCs by assessing changes in cell morphology, cell viability, and gene expression. The hydrogel combination provides a new soft biomaterial platform to study the effect of the OSCAR-collagen interaction on osteoclastogenesis. We anticipate that such hydrogel tools will help design future synthetic platforms to address specific research questions on bone health and disease and inform the development of future strategies for bone regeneration/engineering.

## 2. Results and Discussion

### 2.1. OP1_Hyp, OP1_Pro, and OCol1 Adopt a Collagen Triple Helical Structure

The formation of the collagen triple-helical structure was confirmed by CD spectroscopy (Figure 1). OP1_Hyp, OP1_Pro, and OCol1 showed a band of positive ellipticity with a maximum at ~220 nm and a deep band of negative ellipticity with a minimum at ~198 nm, previously associated with the typical conformation of the collagen triple helix [23,24,25,26]. The triple helical features, in particular the positive ellipticity band, disappeared from the CD spectra when the temperature was increased to 70 °C, while upon cooling to 10 °C, the three spectra almost completely recovered their triple helical characteristics after approximately 30 min (Figure 1A). Refolding of the collagen triple helical structure was much more evident for OCol1, with a CD spectrum practically indistinguishable from the initial one. OP1_Hyp and OP1_Pro showed partial recovery of the collagen structure despite having a much higher molar concentration (210 µM) than OCol1 (12 µM). The faster refolding for OCol1 is likely to be driven by the PfC trimerization domain, which helps in the formation of the triple helical structure [22], and probably also by the longer length of its collagen domain (72 amino acids) compared to the shorter peptides (27 amino acids) (Table 1).

To assess the thermal stability of the collagen peptides and OCol1, denaturation of the collagen triple helix was monitored at 220 nm as a function of a continuously increasing temperature, from 10 °C to 60 °C (OP1_Hyp and OP1_Pro) or to 70 °C (OCol1). Each thermal curve showed a single transition, which typically corresponds to the decrease of ellipticity at 220 nm and the loss of the collagen triple helical structure (Figure 1B). Both OP1_Hyp and OP1_Pro showed a transition from trimer to monomer with *T_m_* values of 36 °C and 27 °C, respectively [24]. These values are consistent with those measured for peptides with similar sequences and lengths [9,13]. Substitution of just one Hyp residue in OP1_Hyp to Pro in OP1_Pro (Table 1) resulted in a significant decrease in thermal stability, as expected from the role of hydroxyproline in the stabilisation of the collagen triple helical structure [12]. For OCol1, a single and sharp transition was observed at a higher temperature, *T_m_* = 60 °C, which may be due to a stabilising effect of the PfC trimerization domain [22].

### 2.2. Both OP1_Hyp and OP1_Pro Bind to OSCAR, but with Different Affinities

Surface plasmon resonance (SPR) [27,28] was used to determine the binding affinity between OP1_Hyp or OP1_Pro and a recombinant human OSCAR-Fc chimera made from the extracellular region of human OSCAR (Asp19-Asn229, NP_573399) and the Fc domain from human IgG1 (Pro100-Lys330, P01857-1). A kinetic analysis was performed with immobilised OSCAR-Fc, OP1_Hyp, and OP1_Pro as analytes. A Langmuir 1:1 model was used to perform the kinetic analysis. This model is based on a simple interaction between ligand and analytes where a 1:1 molar complex is formed at equilibrium, i.e., A + B ⇄ AB. The kinetics of the reaction are determined by the association and dissociation rate constants (*k*_a_ and *k*_d_, respectively), with an overall affinity constant (K_D_) given by K_D_ = *k*_d_*/k_a_* [27].

As shown from the sensorgram association and dissociation phases (Figure 2A), OP1_Hyp showed a progressive binding phase to OSCAR with a slow release at all the concentrations tested. From the Langmuir 1:1 analysis, the affinity constant (K_D_) was determined to be 2.63 µM, suggesting that OP1_Hyp represents a high-affinity binding partner for human OSCAR. This value of K_D_ in the low µM range is consistent with the affinities of similar OSCAR-binding peptides measured from solid-phase assays [9].

On the other hand, the OSCAR-OP1_Pro sensorgrams (Figure 2B) showed that both association and dissociation phases occurred very quickly. For this reason, a slightly weaker interaction can be assumed between human OSCAR and OP1_Pro. As such, a steady-state analysis (SSA) was used to measure the affinity between OSCAR and OP1_Pro (Figure 2C). The SSA is a mathematical and computational technique used to study the long-term behaviour of chemical reactions. In particular, it focuses on the ‘steady state’ or equilibrium of concentrations in systems where reaction rates are balanced and concentrations do not change significantly over time. The SSA led to K_D_ = 19.5 µM, indicating a lower affinity of OSCAR for OP1_Pro compared to OP1_Hyp (K_D_ = 2.63 µM). One possible interpretation of these results is that the OSCAR-binding motif GAOGPAGFA has higher affinity for OSCAR when the proline residue at position 3 is hydroxylated (GAOGPAGFA, as in OP1_Hyp), compared to the non-hydroxylated case (GAPGPAGFA, as in OP1_Pro). Nevertheless, OSCAR binding is not abolished by the replacement of this Hyp residue with Pro. Previous studies have concluded that the Hyp residue at position 3 of the consensus motif GXOGPX’GFX’ is critical for OSCAR binding, but only the replacement with alanine was investigated [5]. Our finding suggests that collagen GXPGPX’GFX’ motifs should still be able to bind OSCAR, albeit with lower affinity.

The differences in thermal stability between OP1_Hyp and OP1_Pro may also play a role. The SPR experiment was conducted at 25 °C, close to the melting temperature of OP1_Pro, which will be partially unfolded. The higher melting temperature of OP1_Hyp indicates that this peptide will be mostly in its triple helical conformation at the temperature of the SPR experiment. Previous work has demonstrated that the triple helical structure is needed for OSCAR binding [9,13], and only longer, more stable collagen-like peptides containing OSCAR binding motifs are able to inhibit osteoclastogenesis in cell culture [13]. Based on this interpretation, the Hyp in the OSCAR-binding motif contributes to a more stable collagen peptide and therefore to a higher affinity for OSCAR.

### 2.3. OSCAR Contributes to Raw 264.7 Cell Differentiation towards OCs

To assess the involvement of OSCAR during osteoclastogenesis, Raw 264.7 cells were cultured on tissue culture plastic (TCP) in the presence of OP1_Hyp or OP1_Pro. As expected, Raw 264.7 cells showed typical OC features when cultured for 5 days in the presence of 100 ng/mL of hRANKL for both OP1_Hyp and OP1_Pro (Figure 3). For instance, treated cells were larger in diameter (average diameter of 273.9 ± 73.4 µm) than untreated ones (average diameter of 38.1 ± 8.8 µm), showing the presence of an actin ring surrounding more than three nuclei (highlighted by the purple colour following TRAP staining). In this case, the effect of the peptides could not be seen due to the high concentration of hRANKL that masked any co-stimulatory effect. It is not clear what the lowest concentration of hRANKL is to induce osteoclastogenesis in Raw 264.7 cells cultured on TCP. Different research groups have used different protocols and concentrations of RANKL for Raw 264.7 cell culture and differentiation, and there is a surprising lack of systematic studies on the effect of RANKL concentration. A study by Song et al. [29] concluded that 30 ng/mL is the lowest optimal concentration of RANKL for osteoclastic cell formation. Thus, we also used suboptimal hRANKL concentrations of 10 ng/mL and 1 ng/ml. Raw 264.7 cells still showed signs of osteoclastogenesis when cultured with 10 ng/mL of hRANKL in the presence of OP1_Hyp or OP1_Pro, but the changes in their morphology were less pronounced. In particular, the effect was more evident when they were cultured on OP1_Hyp, with more cells showing OC-like features such as multinucleation, a hallmark of OC maturation [30], and multiple invaginations of the cell membrane (often referred to as a ruffled border), compared to those cultured on OP1_Pro. This observation is consistent with the lower OSCAR-binding affinity for OP1_Pro. No differentiation was observed when Raw 264.7 cells were cultured with 1 ng/mL of hRANKL. Additionally, OP1_Hyp and OP1_Pro were not able to trigger any differentiation on their own (i.e., with no hRANKL added), as shown by the absence of TRAP staining (Figure 3).

Differentiation to OCs and the presence of any co-stimulatory effect were also assessed by analysing typical OC marker expression, i.e., TRAP and OSCAR, via qRT-PCR (Figure 4). In this experiment, Raw 264.7 cells were treated with three doses of hRANKL: 100, 10, and 1 ng/mL. As expected, an increase in gene expression confirmed an overall involvement of OSCAR during the differentiation of Raw 264.7 cells. Indeed, compared to day 1, the level of TRAP expression was significantly higher after 5 days for cells treated with 100 ng/mL of hRANKL and cultured with either OP1_Hyp or OP1_Pro. Similarly, OSCAR expression was also significantly higher compared to day 1, suggesting that Raw 264.7 cells were potentially interacting with both OP1_Hyp and OP1_Pro through OSCAR. A similar pattern of expression was observed when cells were treated with a lower concentration of hRANKL (10 ng/mL). Indeed, although TRAP expression was lower, OSCAR gene expression was still significantly higher for cells cultured on either OP1_Hyp or OP1_Pro. This suggests not only that there were cells able to differentiate into OCs, as confirmed by the TRAP expression, but also that OSCAR was involved during the differentiation. On the other hand, when cells were treated with the minimal dose of hRANKL tested (0–1 ng/mL), both TRAP and OSCAR genes were either expressed at lower amounts or did not change significantly after five days of culture. Despite the differences in stability and binding affinity observed between OP1_Hyp and OP1_Pro, both peptides showed a positive effect on differentiation when used as a substrate to culture Raw 264.7 cells. It is worth mentioning that the OSCAR-collagen-like peptide equilibrium in solution (as in an SPR experiment) is not necessarily representative of the situation where the collagen-like peptides are coated on TCP. The local saturation of OP1_Pro on the well surface may still be able to provide cell adhesion and differentiation as effectively as OP1_Hyp. For this reason, we think that Raw 264.7 cells were able to grow and differentiate into mature OCs on both OP1_Hyp and OP1_Pro when treated with a high concentration of hRANKL (100 ng/mL). Similarly, differentiation of Raw 264.7 cells was still observed with a lower, suboptimal concentration of hRANKL [29], although the overall effect was much more evident for OP1_Hyp compared to OP1_Pro. To assess the effect of OSCAR as a co-stimulatory effect on OCP differentiation, 10 ng/mL of hRANKL was chosen as the optimal concentration for further studies, as a higher concentration of the cytokine might mask any additional effects.

### 2.4. OSCAR-Modified Hydrogels for OC Culture and Differentiation

To further test the co-stimulatory effect of OSCAR during Raw 264.7 cell differentiation to mature OCs, we used the Hap-decorated Fmoc-FF/S/RGD hydrogels formulated in our previous work (FF/S/RGDH) [20] and modified the gels with 100 µg/mL of either OP1_Hyp, OP1_Pro, or OCol1 (referred to collectively here as collagen-modified hydrogels). As stated earlier, we chose 10 ng/mL of hRANKL as the concentration to trigger Raw 264.7 cell differentiation. Figure 5 shows the analysis of F-actin staining of Raw 264.7 cells cultured on different collagen-modified FF/S/RGDH hydrogels. Cells were cultured for 7 days, and OC-like features, such as dense actin rings and multinucleation, were assessed. Results are presented in comparison with the unmodified FF/S/RGDH hydrogel to assess whether the addition of OSCAR-binding peptides or of OCol1 had a positive or detrimental effect on the differentiation of Raw264.7 cells to OCs. After seven days of culture, Raw 264.7 cells showed an OC-like morphology with a prominent ruffled border, which is surrounded by an F-actin-rich structure (actin ring) (Figure 5). This area contains densely packed podosomes (Figure 5) that generate focal adhesion with the hydrogel matrix.

This is a typical OC feature extensively described in the literature [31,32]. Moreover, the actin ring surrounded more than three nuclei in all the formulations tested, which is usually associated with the morphology of mature OCs [33,34].

Multinucleation was also assessed by counting the number of nuclei/cells. Compared to unmodified FF/S/RGDH, OP1_Hyp-modified and OCol1-modified hydrogels demonstrated a higher proportion of multinucleated cells (3–5 nuclei/cell), with statistically significant differences (*p* = 0.013 and 0.045, respectively). Significant differences were also observed for OP1_Hyp-modified or OCol1-modified hydrogels compared to their OP1_Pro-modified counterparts (*p* = 0.012 and 0.040, respectively), while no significant difference was observed between hydrogels modified with OP1_Hyp and OCol1 (*p* = 0.669) or between the unmodified hydrogel and the OP1_Pro-modified one (*p* = 0.938) (Figure 6A). Raw 264.7 cells cultured on OP1_Hyp-modified hydrogels showed a significant increase in diameter compared to FF/S/RGDH, with a lognormal distribution (µ = 33.85, σ = 1.34) shifted towards larger cell diameters compared to the other hydrogel substrates (µ = 28.76, σ = 1.44 for FF/S/RGDH; µ = 25.33, σ = 1.27 for FF/S/RGDH+OP1_Pro; and µ = 30.82, σ = 1.59 for FF/S/RGDH+OCol1) (Figure 6B). Cells cultured on the different collagen-modified hydrogels showed typical features of mature OCs (e.g., a ruffled border, a dense actin ring, and multiple nuclei per cell). These results are in line with those described by Barrow et al., where OSCAR-binding peptides were able to act as co-stimulatory factors for OC differentiation [5]. Here, although all three collagen-modified hydrogels triggered the differentiation of Raw 264.7 cells, the effect was more evident with OP1_Hyp.

These results agree with our previous observations. Indeed, OP1_Hyp has been shown to interact more efficiently with OSCAR (Figure 2) and to have higher thermal stability (Figure 1B) due to the presence of Hyp in the GAOGPAGFA OSCAR-binding sequence. For this reason, Raw 264.7 cells may have a stronger affinity for OP1_Hyp, and therefore the differentiation effect (as shown by OC features in Figure 5 and Figure 6) may be more pronounced. Additionally, the viability of Raw 264.7 cells cultured on the collagen-modified hydrogels was analysed via a LIVE/DEAD assay (Figure 7).

In our previous work, we demonstrated high viability over 7 days for Raw 264.7 cells cultured on the Hap-decorated Fmoc-FF/S/RGD hydrogels [20]. Here, we wanted to test the effect of collagen-like peptide-loaded hydrogels on cell growth. As shown in Figure 7, both at day 3 and day 7 of culture, the majority of cultured cells were viable (green dye), starting to form small clusters over time. High viability and clustering effects highlighted how collagen-modified hydrogels, especially for OP1_Hyp and OCol1, were able to maintain the high viability of Raw 264.7 cells as well as support their differentiation towards mature OCs over time.

To confirm the gene expression of typical OC markers as well as the involvement of OSCAR during the osteoclastogenesis of Raw 264.7 cells were cultured on the collagen-modified hydrogels, and qRT-PCR was used. Expression of TRAP and OSCAR was analysed on days 3 and 7. When cells were cultured on OP1_Hyp- and OCol1-modified hydrogels, TRAP expression increased significantly (*p* < 0.001) from day 3 to day 7, a change that was not observed on OP1_Pro-modified hydrogels (Figure 8A). Significant increases in OSCAR expression were observed on all the collagen-modified hydrogels from day 3 to day 7 (Figure 8B). Recently, Lampiasi et al. [35] have studied the expression levels of OSCAR and TRAP in Raw 264.7 cells were treated with 50 ng/mL of RANKL only. In their study, OSCAR and TRAP levels increased significantly between days 1 and 2 of culture before decreasing sharply on days 3 and 4 [35]. In our study, the expression levels of OSCAR and TRAP increased significantly between day 3 and day 7 of culture, especially for OP1_Hyp and OCol1. Moreover, in our experiment, a suboptimal concentration of RANKL (10 ng/mL) was used. Coupled together, these results reveal a positive effect on osteoclastogenesis induced by the addition of OSCAR-binding collagen-like peptides and proteins in culture, possibly as they better mimic the composition of bone’s ECM and provide the right cell-matrix biochemical cues for OCs to progress in their differentiation.

According to these results, OP1_Hyp and OCol1 incorporated within the FF/S/RGDH hydrogel can efficiently bind OSCAR, and therefore, after a minimal stimulation with hRANKL (10 ng/mL), Raw 264.7 cells can be committed to osteoclastogenesis, as shown by F-actin staining and nuclei distribution. The reduced interaction with OP1_Pro could be related to its lower thermal stability and lack of a collagen triple helical conformation at the temperature conditions used for cell culture on the hydrogels (37 °C). Taking into consideration the changes in Raw 264.7 cell morphology, the increase of multinucleation, and the overexpression of OC-typical markers, it can be concluded that OP1_Hyp and OCol1 have a positive effect on osteoclastogenesis when incorporated into the Hap-decorated Fmoc-FF/S/RGD hydrogels.

## 3. Materials and Methods

### 3.1. OSCAR-Binding Peptides and Recombinant Protein

Two collagen-like peptides containing an OSCAR binding motif [9] with Hyp (GAOGPAGFA) or Pro (GAPGPAGFA) at the 3rd position were synthesised by GenScript Biotech (Oxford, UK) and provided in lyophilised form. The two peptides are hereafter referred to as OP1_Hyp (batch No. U0503EK140_1) and OP1_Pro (batch No. U0503EK140_3), respectively. Purity was assessed by HPLC to be 95.3% for OP1_Hyp and 95.0% for OP1_Pro. A recombinant collagen-like protein (called OCol1 here) was designed with the potential OSCAR-binding sequence GPPGPQGFQ and a domain architecture identical to that of the recombinant protein DCol1 [13]. Table 1 summarises the collagen-like peptides OP1_Hyp and OP1_Pro and the recombinant collagen-like protein OCol1 used in this work. OCol1 gene synthesis, subcloning using a pET28c(+) vector, transforming into competent *Escherichia coli* BL21 DE23 cells, and recombinant protein expression and purification by immobilised metal affinity chromatography were outsourced to GenScript Biotech (Oxford, UK). The purity of OCol1 (Batch No. U6699HA190-1_OCol1) was estimated at 85% via SDS-PAGE under reducing conditions. Purified OCol1 was received in a storage buffer (50 mM Tris-HCl, 150 mM NaCl, 10% glycerol, pH 8.0) and stored at −80 °C upon arrival.

### 3.2. Circular Dichroism (CD) Analysis of Secondary Structure and Thermal Stability

Secondary structures of OP1_Hyp, OP1_Pro, and OCol1 were analysed by circular dichroism (CD) spectroscopy using a Jasco^®^ J-810 spectropolarimeter (UCI, Irvine, CA, USA) equipped with a Peltier temperature controller. OP1_Hyp and OP1_Pro were resuspended up to a concentration of 0.5 mg/mL in a CD transparent buffer (10 mM K_2_HPO_4_, 10 mM KH_2_PO_4_, 150 mM KF, pH 7.4, CD Buffer) [15] and kept for 24 h at 4 °C to allow for the formation of the collagen triple helical structure. OCol1 was exchanged to CD Buffer using a PD 10 disposable desalting column (GE17-0851-01, Merck, Feltham, UK). Briefly, the column was equilibrated with five column volumes of CD Buffer before the addition of OCol1 to the storage buffer. After flow-through collection, 0.5 mL fractions were collected into 1.5 mL Eppendorf tubes. The OCol1 elution fractions were quickly analysed by CD, and the one with the highest concentration (0.2 mg/mL) was used. CD spectra were measured between 190 nm and 260 nm at 10 °C using a 1 mm path-length CD-matched quartz cuvette (Starna Scientific, Ilford, UK). Data were collected every 0.2 nm with a 1 nm bandwidth. Spectral baselines were corrected by subtracting the spectrum of CD Buffer (blank) collected under the same conditions. Additionally, the thermal stability of OP1_Hyp, OP1_Pro, and OCol1 was analysed. Thermal transition profiles were recorded between 10 °C and 60 °C (OP1_Hyp and OP1_Pro) or 70 °C (OCol1) at 222 nm with a data pitch of 0.2 °C, a bandwidth of 1 nm, a detector response time of 32 s, and a temperature slope of 1 °C/min. Samples were cooled back to 10 °C after the different transitions had been completed, and final spectra were recorded at that temperature. Ellipticity in millidegrees was converted to mean residue molar ellipticity Θ (degree cm^2^ dmol^−1^) by normalising for the number of residues on each peptide or protein [23].

### 3.3. Surface Plasmon Resonance (SPR) Analysis of the Peptide-OSCAR Interaction

SPR analyses were carried out on a Biacore™ T200 (Cytiva) using a Series S CM5 sensor chip with an assay running buffer of 10 mM PBS, 150 mM NaCl, 0.05% Tween20, pH 7.4. Reference and active flow cells were activated by injecting 70 μL of a 1:1 mixture of 0.2 M 1-ethyl-3-(3-dimethylaminopropyl)-carbodiimide (EDC) and 0.1 M N-hydroxysuccinimide (NHS) at 10 µL/min. Recombinant human OSCAR-Fc chimera (R&D Systems, Bio-Techne, Minneapolis, MN, USA, 9955-OS) was immobilised onto the active flow cell at 20 µg/mL in 10 mM sodium acetate pH 5.0 at 10 µL/min to 6880 response units (RU). Reference and active flow cells were blocked using 1 M ethanolamine, pH 8, at a flow rate of 10 µL/min. For kinetic and equilibrium binding analysis, OP1_Hyp and OP1_Pro peptides were flowed at 25 °C active and reference flow cells at 30 µL/min in assay running buffer at peptide concentrations of 5000—0 nM (OP1_Hyp) and 40,000—0 nM (OP1_Pro), using a dilution series. Surfaces were regenerated with a 10 s injection of 0.1 M glycine, pH 2.0. Data sets were analysed with the built-in BiaEvaluation software (version 4.1). The models used for analysis were the Langmuir 1:1 binding model and the steady-state analysis model. Data are presented as the mean and standard error of the mean (SEM) of triplicate runs.

### 3.4. Raw 264.7 Cell Adhesion Assay

Raw 264.7 cells were cultured and passaged as previously described [20]. Cell adhesion to OP1_Hyp and OP1_Pro was assessed as shown by Barrow et al. [5]. Briefly, 48-well plates were coated with 10 µg/mL of OSCAR-binding peptides overnight at 4 °C. The excess peptide was then removed by washing with PBS before adding 1% BSA to block non-specific binding sites for 1 h at 25 °C. Raw 264.7 cells were then resuspended at a concentration of 2 × 10^5^ cells/mL and added to the well. Cells were cultured with different concentrations of human RANKL (hRANKL) (catalogue number: 310-01 from Peprotech, London, UK) (0–100 ng/mL) for up to 5 days to assess the contribution of the OSCAR-binding peptides OP1_Hyp and OP1_Pro to the differentiation of Raw 264.7 cells to mature OCs.

### 3.5. Peptides and Peptide Hydrogel Fabrication

Fmoc-FF/S (Fmoc-diphenylalanine/serine, 1:1 molar ratio) and Fmoc-FF/S/RGD (1:0.5:0.5 molar ratio) peptides were provided by Biogelx Ltd., Edinburgh, UK. Hydroxyapatite (Hap) nanopowder (formula: Ca_10_(PO_4_)_6_(OH)_2_, with average particle size <200 nm) was purchased from Sigma-Aldrich (Dorset, UK, product code: 677418) and used as received. Hydrogels made up of Fmoc-based peptides and hydroxyapatite were fabricated as described previously [20]. Hap-decorated Fmoc-FF/S/RGD hydrogels (FF/S/RGDH) were tested in combination with OP1_Hyp, OP1_Pro, and OCol1 to evaluate the contribution of OSCAR as a co-stimulatory pathway for OCP differentiation. OP1_Hyp, OP1_Pro, and OCol1 were added to the FF/S/RGDH hydrogels using our incorporation protocol [13]. Briefly, OP1_Hyp, OP1_Pro, and OCol1 were resuspended in PBS up to 100 µg/mL and dispensed on the surface of the pre-gel. Hydrogels were left to crosslink overnight at 4 °C.

### 3.6. Cell Culture and Assessment of Viability and Morphology

The day after hydrogel fabrication, 2 mL of Raw 264.7 cells (4 × 10^5^ cells/mL) were seeded on top of FF/S/RGDH hydrogels containing the collagen-like OSCAR-binding peptides (OP1_Hyp, OP1_Pro) or recombinant collagen-like protein (OCol1). Furthermore, 10 ng/mL of hRANKL was added to the cell media to differentiate the Raw 264.7 cells (OCPs) from OCs. Cells were cultured for up to 7 days at 37 °C and 5% CO_2_ with Dulbecco’s Modified Eagle Medium (DMEM) containing 10% (*v*/*v*) foetal bovine serum (FBS) and 5% (*v*/*v*) Penicillin-Streptomycin-Amphotericin antibiotic mixture (PSA, 100 units/mL penicillin, 100 µg/mL streptomycin, 0.25 µg/mL amphotericin) (Sigma-Aldrich, Dorset, UK). The culture medium was replenished every other day. After 7 days of culture, cell viability on the peptide- or protein-modified FF/S/RGDH hydrogels was evaluated by using a LIVE/DEAD assay (ThermoFisher Scientific, Swindon, UK). Briefly, 600 µL of the assay solution containing 4 µM Ethidium homoDimer-1 (EthD-1) and 2 µM calcein-AcetoxyMethyl (calcein-AM) in PBS were pipetted onto the cell-hydrogel constructs. After 30 min of incubation, cells were washed 3 times in PBS and imaged using a Nikon Eclipse 50i fluorescence microscope (Tokyo, Japan) (emission wavelengths: green channel for live cells: 515 nm; red channel for dead cells: 635 nm; excitation wavelength: 495 nm). Cell differentiation was also assessed by measuring the differences in morphology between modified and unmodified hydrogels. In particular, cell diameter, multinucleation, and the presence of typical OC features were observed by using F-actin staining [20]. The number of nuclei per cell was counted and plotted as violin plots, while cell diameters were plotted using a percentage frequency distribution and fitted with a lognormal distribution curve.

### 3.7. Tartrate-Resistant Acid Phosphatase (TRAP) Staining

Successful differentiation of OCs was demonstrated by tartrate-resistant acid phosphatase (TRAP) staining using a Leukocyte Acid Phosphatase (TRAP) Kit (Sigma-Aldrich, St. Louis, MO, USA). After 5 days of post-culture, Raw 264.7 cells were fixed in 4% PFA for 15 min at room temperature. TRAP was stained following the manufacturer’s protocol and evaluated by light microscopy (ThermoFisher, Swindon, UK). Multinucleated TRAP+ cells containing more than three nuclei were scored as mature OCs.

### 3.8. Gene Expression

Gene expression of OC markers was evaluated as previously described [20]. A StepOne™ Real-Time PCR System (Applied Biosystems, Warrington, UK) was used, with TaqMan probes and a universal PCR Master Mix (Life Technologies, Carlsbad, CA, USA, 4304437) in a total volume of 10 µL. The TaqMan probes for TRAP and OSCAR were Mm00475698_m1 and Mm01338227_g1, respectively. Data were analysed using the 2^−∆Ct^ method and normalised to the endogenous housekeeping gene GAPDH (Mm99999915_g1). Experiments were performed in triplicate.

### 3.9. Statistical Analysis

All quantitative values are presented as mean ± standard deviation. All experiments were performed using at least three replicates. The data were plotted using Origin 2019b. The numbers of nuclei per cell were shown as violin plots (Figure 6A) and compared using the Mann–Whitney U non-parametric test (as the numbers of nuclei per cell are not normally distributed). All the other tests were compared using an unpaired *t* test, unless stated otherwise. One level of significance was used: *p* < 0.05 (*).

## 4. Conclusions

In this paper, two collagen-like peptides (OP1_Hyp and OP1_Pro) and one recombinant collagen-like protein (OCol1) containing OSCAR-binding motifs have been combined with Hap-modified Fmoc-FF/S/RGD hydrogels to test in vitro the effect of collagen-OSCAR interactions during osteoclastogenesis. All the collagen-like molecules adopted the collagen triple helical conformation required for OSCAR binding [9,13], as shown by CD spectroscopy, while their thermal stability varied depending on their sequence (Tm values of 36 °C, 27 °C, and 60 °C for OP1_Hyp, OP1_Pro, and OCol1, respectively). In particular, the replacement of a single Hyp with a Pro residue significantly decreased the stability of OP1_Pro, consistent with the role of prolyl hydroxylation on the overall stability of collagen triple helices [12]. SPR results confirmed a higher affinity of OP1_Hyp for OSCAR compared to OP1_Pro (K_D_ = 2.63 µM vs. 19.5 µM), consistent with its higher thermal stability and preservation of the triple helix. When cultured on Fmoc-FF/S/RGD/Hap hydrogels modified by either OP1_Hyp, OP1_Pro, or OCol1, Raw 264.7 cells showed high viability and improved OC-like features compared to the unmodified hydrogels, with OP1_Hyp and OCol1 showing greater bioactivity (i.e., higher nuclei/cell distribution, increased cell diameter). Despite the high biocompatibility and typical morphology, only OP1_Hyp- and OCol1-modified hydrogels caused a significant upregulation of typical OC markers, such as TRAP and OSCAR. Taken together, these results confirm that the incorporation of OSCAR-binding collagen-like peptides or proteins into the Fmoc-FF/S/RGD/Hap hydrogel promoted enhanced osteoclast differentiation with no detrimental effect on Raw 264.7 cell survival. We envisage that our in vitro hydrogel system modified with collagen-like OSCAR-binding motifs will serve as a suitable soft composite scaffold to address specific questions about osteoclastogenesis.

## Figures and Tables

**Figure 1 ijms-25-00445-f001:**
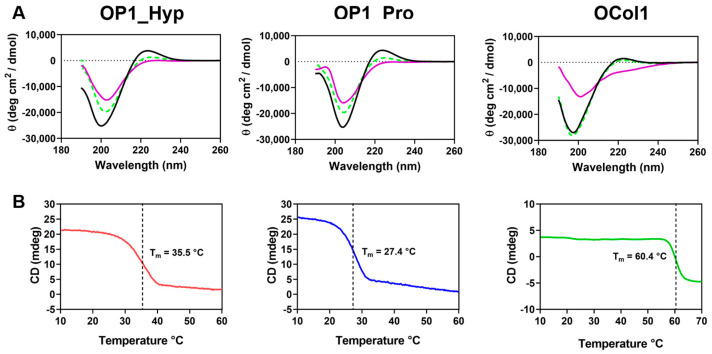
(**A**) CD spectra at 10 °C (black line), 70 °C (purple line), and 10 °C after denaturation and refolding (green dashed line) for OP1_Hyp, OP1_Pro, and OCol1. The vertical axis measures mean residue ellipticity Θ in degrees cm^2^ dmol^−1^. CD data were collected between 190 and 260 nm. (**B**) Thermal denaturation of OP1_Hyp (red line), OP1_Pro (blue line), and OCol1 (green line) was monitored by the change of CD at 220 nm as a function of increasing temperature between 10 °C and 70 °C at a heating rate of 1 °C/min. Concentrations were 0.5 mg/mL for OP1_Hyp and OP1_Pro (210 µM) and 0.2 mg/mL for OCol1 (12 µM), in CD Buffer pH 7.4.

**Figure 2 ijms-25-00445-f002:**
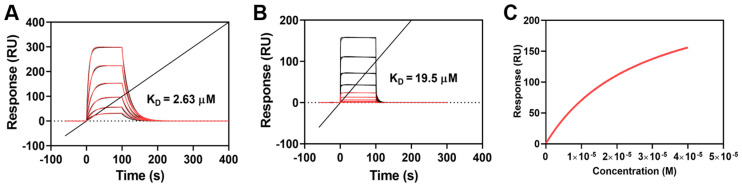
SPR multi-cycle kinetic analysis of the interaction of OSCAR-Fc with OP1_Hyp and OP1_Pro. (**A**,**B**) Sensorgrams of OSCAR-OP1_Hyp and OSCAR-OP1_Pro were analysed using the Langmuir 1:1 model. Relative responses are shown in red, with theoretical fit in black. (**C**) OP1_Pro steady state analysis as a measure of affinity vs. OP1_Pro concentration.

**Figure 3 ijms-25-00445-f003:**
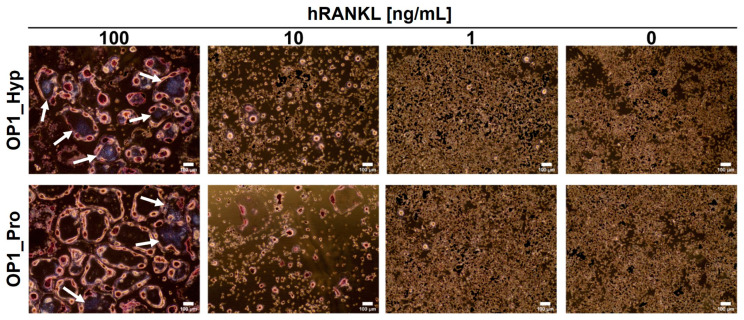
TRAP straining of Raw 264.7 cells were cultured for 5 days on OP1_Hyp/OP1_Pro-coated 24-well plates. Different concentrations of hRANKL (0–100 ng/mL) were used. Cells were fixed and stained to detect TRAP activity. TRAP-positive cells (violet-labelled, pointed by white arrows) with at least three nuclei were considered osteoclasts. Scale bar = 100 µm, 10× magnification.

**Figure 4 ijms-25-00445-f004:**
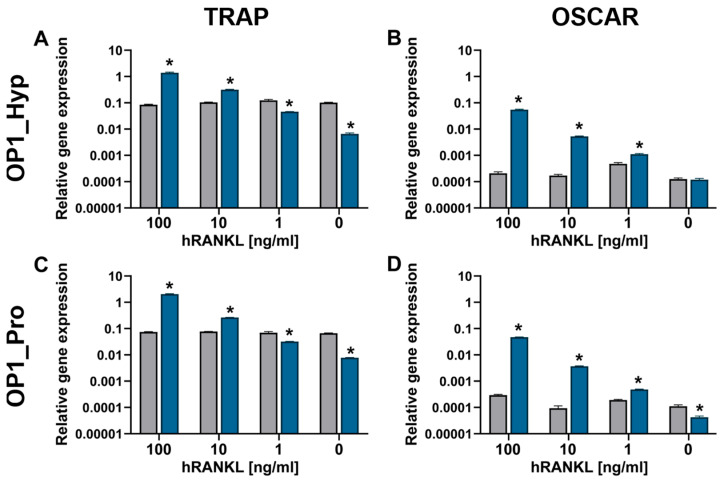
Gene expression of TRAP (**A**,**C**) and OSCAR (**B**,**D**) relative to GAPDH by Raw 264.7 cells (*n* = 3) after 24 h (grey bars) and five days (blue bars) cultured on OP1_Hyp (**A**,**B**) and OP1_Pro (**C**,**D**) with different concentrations of hRANKL (0–100 ng/mL). All data are shown as mean ± SD; * *p* < 0.05.

**Figure 5 ijms-25-00445-f005:**
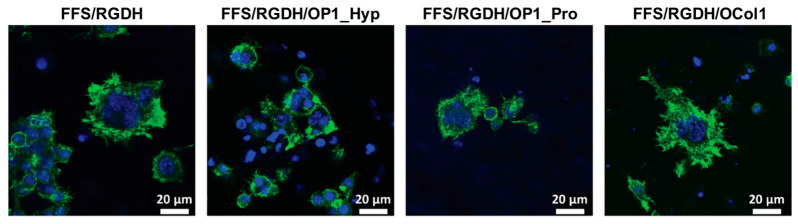
Raw 264.7 cell morphology. Fluorescence images of Raw 264.7 cells after seven days of culture on FF/S/RGDH hydrogels, either unmodified or with incorporated OP1_Hyp, OP1_Pro, or OCol1, were treated with 10 ng/mL of hRANKL (green: F-actin, Alexa Fluor 488 Phalloidin; blue: Nuclei, Hoechst 33342; scale bars 20 μm).

**Figure 6 ijms-25-00445-f006:**
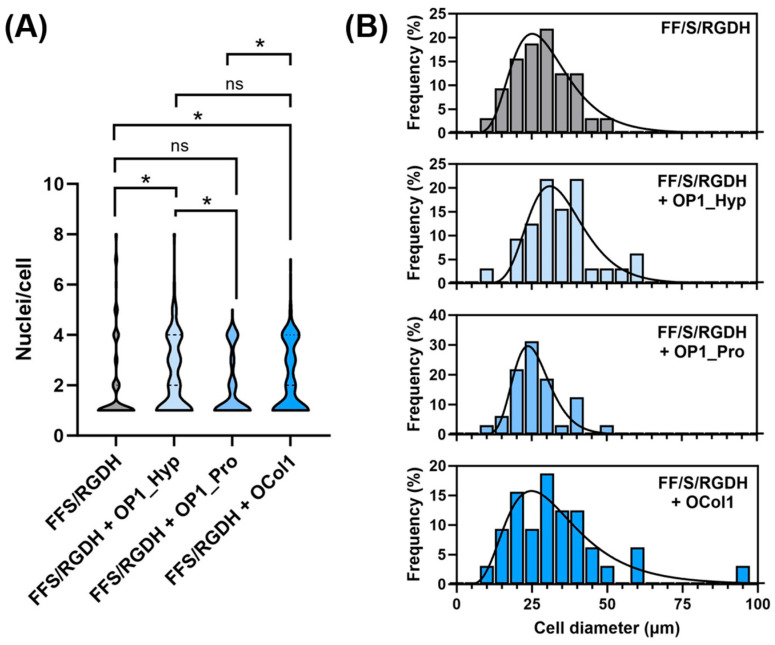
Raw 264.7 multinucleation and cell size. (**A**) Analysis of multinucleation by a violin distribution plot. Data shown as the number of nuclei per cell, *n* = 100; statistical comparison performed via the Mann–Whitney U non-parametric test, * *p* < 0.05, ns: not significant. (**B**) Distribution of cell diameter. Data shown as percentage frequency distribution, *n* = 30, fitted with a lognormal distribution fit.

**Figure 7 ijms-25-00445-f007:**
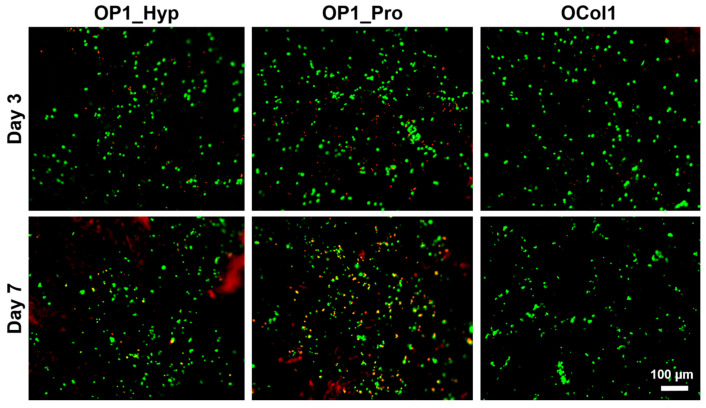
Analysis of the viability of Raw 264.7 cells cultured on collagen-modified FF/S/RGDH hydrogels at day 3 and day 7 using a LIVE/DEAD assay (green: viable cells, calcein AM; red: dead cells, ethidium homodimer-1). Scale bar = 100 µm, 10× magnification.

**Figure 8 ijms-25-00445-f008:**
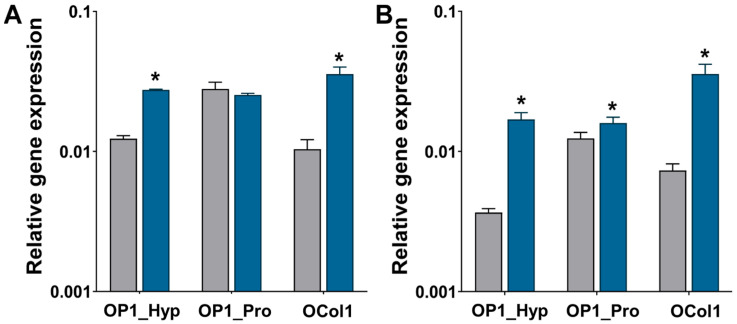
Gene expression of TRAP (**A**) and OSCAR (**B**) relative to GAPDH by Raw 264.7 cells were cultured on collagen-modified hydrogels. Gene expression was assessed for Raw 264.7 cells (*n* = 3) after three (grey bars) and seven (blue bars) days were cultured on collagen-modified hydrogels and treated with a 10 ng/mL concentration of hRANKL. All data are shown as mean ± SD; * *p* < 0.05.

**Table 1 ijms-25-00445-t001:** Collagen-like peptides OP1_Hyp and OP1_Pro and recombinant collagen-like protein OCol1 were used to modify Hap-decorated Fmoc-FF/S/RGD (FF/S/RGDH) hydrogels. Amino acid sequences are shown in standard single-letter code, plus O for 4-hydroxyproline. Known or potential OSCAR binding sites are shown in bold underlined green. The PfC trimerization domain (see text and [22]) is shown in bold blue type. Ac-, N-terminal acetylation; -NH_2_, C-terminal amidation. Theoretical molecular weights (Mw) were calculated from the amino acid sequences.

Molecule	Sequence	Amino Acids	Monomer Mw (Da)	Trimer Mw (Da)
OP1_Hyp	Ac-GPOGPOGPO**GAOGPAGFA**GPOGPOGPO-NH2	27	2404.56	7213.68
OP1_Pro	Ac-GPOGPOGPO**GAPGPAGFA**GPOGPOGPO-NH2	27	2388.56	7165.68
OCol1	MGSHHHHHHSGLVPRGSGPPGPPGPQGPAGPRGEPGPAGPKGEPGPA**GPPGPQGFQ**GPPGPQGPAGPIGPKGEPGPIGPQGPKGDPGET**QIRFRLGPASIIETNSNGWFPDTDGALITGLTFLAPKDATRVQGFFQHLQVRFGDGPWQDVKGLDEVGSDTGRTGE**	165	16,625.34	49,876.02

## Data Availability

This study is adapted from and reports on data included in the thesis of Mattia Vitale submitted to the University of Manchester as a requirement to obtain the degree of Doctor of Philosophy (Ph.D.). An electronic version of the thesis (eThesis) is available from the University of Manchester Library institutional repository at the following website: https://research.manchester.ac.uk/en/studentTheses/customised-hydrogel-matrices-for-osteoclast-differentiation-and-c (accessed on 1 November 2023).

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
