# Peer review of "Collagen-like Osteoclast-Associated Receptor (OSCAR)-Binding Motifs Show a Co-Stimulatory Effect on Osteoclastogenesis in a Peptide Hydrogel System"

_ijms, 2023, doi:10.3390/ijms25010445_

Round 1

Reviewer 1 Report

Comments and Suggestions for Authors

In this research the authors designed two OSCAR-binding peptides and one recombinant OSCAR binding protein and combined them with peptide-based hydrogels to study their effect on osteoclastogenesis. This is a very interesting topic however there are some questions need to be addressed first before it's publication.

1, from line 42-44. the transfer from OB research to OC research of OSCAR is too stiff. It would be better if use a softer way to introduce OC research here. 

2, line 84, what is the advantage of your designed hydrogels?

3, line193, why use hRANKL here to treat mouse raw cells. Have the author tested about the dose influence of mouse RANKL?

4, line 216, how about the TRAP staining of 24 well plates alone without any peptide coating?

5, line 290, Quantification data is inconstant with figure A. i see some cells are also very big in figure A.

6, line 300, how about the cells growth on hydrogel alone without the peptide loading?

7, line 447-449, can you add more details of how you fictionalized hydrogel with peptide? And how about the modification efficiency?

Reviewer 2 Report

Comments and Suggestions for Authors

1. Figure 3: The current images did not show very clear TRAP-positive stain (purple cells with multi-nuclei). Also, the quantification of osteoclasts should be provided.

2. Figure 5: From the data there seems to be no significant difference between the control material (FFS/RGDH) and the OSCAR-binding motif added materials. These data could not support the conclusion that the OSCAR-binding motif can actually improve osteoclastogenesis.

3. More data are required to demonstrate the role of OSCAR-binding motif in osteoclastogenesis. It is suggested to provide more solid data on the difference between the control hydrogel and the OSCAR-binding motif added hydrogels, including the differences on F-actin ring formation (the data in Figure 5 actually showed no difference) and the expression of osteoclastogenesis markers at the protein and mRNA levels. More solid data regarding the material biocompatibility should be provided, such as cell proliferation and attachment/morphology on the hydrogel.    

Round 2

Reviewer 1 Report

Comments and Suggestions for Authors

The author answered all the questions point to point. But there are something needs to improve before it can be published.

1, Figure3. The background is too dark in the images. Therefore it's hard to see your data very clearly, except the 100ng/mL. 

2, Figure5. The cell numbers and distribution in the images are very ununiform. Therefore, the images and quantification data is hard to compare and looks inconsistent.

3, The conclusion part looks like discussion. Generally conclusion is short and concise, and there is no citation in conclusion. The author can isolate section 2 as results, make section 3 as discussion, and rewrite conclusion.
